# Meta-Analysis of Microarray Data and Their Utility in Dissecting the Mapped QTLs for Heat Acclimation in Rice

**DOI:** 10.3390/plants12081697

**Published:** 2023-04-18

**Authors:** Bablee Kumari Singh, Sureshkumar Venkadesan, M. K. Ramkumar, P. S. Shanmugavadivel, Bipratip Dutta, Chandra Prakash, Madan Pal, Amolkumar U. Solanke, Anil Rai, Nagendra Kumar Singh, Trilochan Mohapatra, Amitha Mithra Sevanthi

**Affiliations:** 1ICAR-National Institute for Plant Biotechnology, Pusa Campus, New Delhi 110012, India; 2PG School, ICAR-Indian Agricultural Research Institute, Pusa Campus, New Delhi 110012, India; 3Division of Plant Biotechnology, ICAR-Indian Institute of Pulses Research, Kanpur 208024, India; 4Division of Plant Physiology, ICAR-Indian Agricultural Research Institute, New Delhi 110012, India; 5ICAR-Indian Agricultural Statistics Research Institute, Pusa Campus, New Delhi 110012, India; 6Indian Council of Agricultural Research, Krishi Bhawan, New Delhi 110001, India

**Keywords:** databasefor heat stress tolerance, stress-responsive genes, candidate genes, allele mining, QTL analysis, http://14.139.229.201/RiceMetaSys/HRG

## Abstract

In the current global warming scenario, it is imperative to develop crops with improved heat tolerance or acclimation, for which knowledge of major heat stress-tolerant genes or genomic regions is a prerequisite. Though several quantitative trait loci (QTLs) for heat tolerance have been mapped in rice, candidate genes from these QTLs have not been reported yet. The meta-analysis of microarray datasets for heat stress in rice can give us a better genomic resource for the dissection of QTLs and the identification of major candidate genes for heat stress tolerance. In the present study, a database, RiceMetaSys-H, comprising 4227 heat stress-responsive genes (HRGs), was created using seven publicly available microarray datasets. This included in-house-generated microarray datasets of Nagina 22 (N22) and IR64 subjected to 8 days of heat stress. The database has provisions for searching the HRGs through genotypes, growth stages, tissues, and physical intervals in the genome, as well as Locus IDs, which provide complete information on the HRGs with their annotations and fold changes, along with the experimental material used for the analysis. The up-regulation of genes involved in hormone biosynthesis and signalling, sugar metabolism, carbon fixation, and the ROS pathway were found to be the key mechanisms of enhanced heat tolerance. Integrating variant and expression analysis, the database was used for the dissection of the major effect of QTLs on chromosomes 4, 5, and 9 from the IR64/N22 mapping population. Out of the 18, 54, and 62 genes in these three QTLs, 5, 15, and 12 genes harboured non-synonymous substitutions. Fifty-seven interacting genes of the selected QTLs were identified by a network analysis of the HRGs in the QTL regions. Variant analysis revealed that the proportion of unique amino acid substitutions (between N22/IR64) in the QTL-specific genes was much higher than the common substitutions, i.e., 2.58:0.88 (2.93-fold), compared to the network genes at a 0.88:0.67 (1.313-fold) ratio. An expression analysis of these 89 genes showed 43 DEGs between IR64/N22. By integrating the expression profiles, allelic variations, and the database, four robust candidates (LOC_Os05g43870, LOC_Os09g27830, LOC_Os09g27650, andLOC_Os09g28000) for enhanced heat stress tolerance were identified. The database thus developed in rice can be used in breeding to combat high-temperature stress.

## 1. Introduction

The global average temperature increased by 0.5 °C in the 20th century owing to the accumulation of greenhouse gases such as carbon dioxide, methane, and nitrous oxide. At the current rate of global warming, the increase in mean temperature is expected to reach more than double the prescribed limit of 1.5 °C [1,2]. The current warming trends have already begun to impact different agricultural crop production systems [3]. If global warming continues at its current rate, severe high-temperature events could become the norm by the end of the twenty-first century, resulting in a decrease in overall crop yields. On average, this would reduce the global production of staple food crops by 6.0% for wheat, 3.2% for rice, 7.4% for maize, and 3.1% for soybeans provided some adaptive measures are followed [4,5]. The grain yield of rice, one of the major staple food crops of humankind, is estimated to be reduced by 41% due to high-temperature stress by the end of the 21st century if no adaptation measures are followed [6]. Besides the adverse effect of heat stress on crop productivity, heat stress can also deteriorate grain quality, causing substantial economic losses [7]. Thus, it is important to accelerate the identification, characterisation, and transfer of heat stress-tolerant genes, at least in major food crops [8].

In rice, heat tolerance studies have primarily focused on the reproductive stage, owing to its high sensitivity as well as its immediate relevance to grain yield and quality [9,10,11,12,13,14]. Extreme temperatures hamper the processes of anther dehiscence, pollen germination, and pollination, ultimately leading to spikelet sterility [15,16]. One of the best ways to sustain rice productivity is to develop heat-tolerant varieties. However, heat stress tolerance is a multi-genic trait, and the candidate genes for heat tolerance are poorly known [13,17,18]. The few known putative candidate genes belong to the classes of transcription factors (e.g., heat shock factors, HSFs), chaperone proteins (alpha-crystalline family HSP20 and DNAJ homologue HSP), protein kinases, phospholipases, proteases, and proteins associated with defence and disease resistance, as well as several other novel proteins that are currently annotated as expressed and hypothetical proteins [19].

Most of the commercially elite and popular rice varieties grow well at an optimum temperature range of 25–35 °C [20]. Temperatures below 33 °C have no adverse influence on spikelet fertility [21]. Hence, the climate modelling and experimental heat stress studies in rice have used temperatures ranging from 33 to 38 °C at the post-booting phase to estimate the effect of increasing temperature on rice grain yield and quality [22,23,24]. Further, rice is reported to show increased spikelet sterility at temperatures beyond 37 °C, hence, most of the studies that identify or characterise genes for heat stress have maintained 38–42 °C during heat stress [9,10,18,25] (Table 1). Rice flowering, the most sensitive period for heat stress incidents, occurs, on average, over a period of six days, between 10 a.m. to 12 p.m., in most of the *O. sativa* genotypes except *O. glaberrima*, in which it is completed by 9 a.m. [21,26]. Hence, most of the studies on the mapping of genes and quantitative trait loci (QTLs) for heat stress tolerance in rice have imposed high-temperature stress for a period of 5–10 days during the booting to early grain formation time [18,25].

In the context of a highly deteriorating global warming scenario, it is imperative to develop crops with improved heat tolerance by means of either molecular breeding or genetic engineering [27,28,29]. Breeding strategies for improving heat tolerance in rice require suitable heat-tolerant donors and a thorough understanding of the heat tolerance mechanisms. Rice genetic resources that are resistant to high temperatures have been identified in both Japonica and Indica subspecies [30]. Out of these, Nagina 22 (N22), *aus* rice variety, and Giza178 are the most heat-tolerant rice varieties [10]; N22 has been extensively utilised as the best heat-tolerant donor in breeding [25,26]. Recently, NL44, a NERICA (NewRIce for AfriCA) rice genotype, has been reported as a heat-tolerant variety [31]. So far, several QTLs for heat tolerance have been mapped in rice utilising different mapping populations targeting different growth stages, e.g., seedling [13,32], booting [33], and flowering [9,10,34,35,36,37,38]. However, the confirmation and fine mapping of these identified QTLs have not been reported yet, except for the study by Ye et al. [39].

One of the most effective and proven means for dissecting complex traits is by coupling the genetic mapping and genome-wide transcriptome profiling of the parental genotypes, which can help to narrow down the candidate gene(s) underlying the functional polymorphism in the QTL regions. This has been demonstrated as one of the major approaches for dissecting any trait of interest at the molecular level in model organisms such as Drosophila and rice [40,41] and has now become a standard approach in the modern genomic era, as evident from several QTL dissection studies [42]. Understanding crop responses at separate ‘omics’ or organisational levels named, genes (genomics), proteins (proteomics), and metabolites (metabolomics), with respect to specific physiology, will augment the meaning and quality of the derived biological information, which, in turn, will help us to develop stress-resilient cultivars. When a large number of genes from various biological materials are implicated in the expression of a trait, meta-analysis is a cost-effective method for identifying the robust candidate gene(s) required for trait improvement through breeding. An approach such as this seeks to identify statistically robust candidate gene(s) from existing data, such as expression microarray datasets.

With the accumulation of a large number of publicly available genome-wide expression datasets, various databases for rice have been developed, each focusing on a specific trait or multiple traits. Some examples include huge databases such as RiceXpro and Gene Investigator and the smaller but trait-specific ones such as RiceMetaSysA for abiotic stresses (drought and salinity stress) and RiceMetaSysB for biotic stresses (rice blast and bacterial leaf blight). In the latter, the genome-wide microarray datasets from a large number of genetic backgrounds for specific stresses have been combined by meta-analysis, and the stress-responsive genes have been made available in the form of databases for their easy retrieval and utilisation [41,43]. They provide detailed information on stress-responsive genes, with relatively simple and straightforward search options, and thus can aid users in the identification of robust candidate genes for a specific stress. Such gene databases can be a valuable source of genes for trait improvement.

Here, we report a database for heat-responsive genes in rice, as identified through the meta-analysis of microarray data under heat stress, from seven different rice cultivars. The meta-analysis includes the dataset (GSE136746) generated in our laboratory from two cultivars contrasting in heat stress response, N22 and IR64. The detailed results of the in-house generated genome-wide expression data and the development of the database are presented in the first two sections of the results. The last section describes the utility of our database by dissecting the three major QTL regions earlier identified for heat stress tolerance in the N22/IR64 population [25]. For this, we carried out allele mining and a gene expression analysis of the QTL-specific genes and their interacting partner genes in N22 and IR64. By integrating these data with the database results, we have proposed a few robust candidate genes that could be playing a major role in conferring heat stress tolerance to rice.

## 2. Materials and Methods

### 2.1. Plant Material, Heat Stress Treatment, and Sampling

A pair of rice cultivars, N22 and IR64, contrasting in their responses to heat stress, was used in both the microarray and gene expression profiling studies. For the collection of panicle tissues for microarray analysis, the methodology described by Shanmugavadivel et al. [25] in the heat stress QTL mapping study was followed. In brief, the plants were grown in the high-precision automated heat stress phenotyping facility at the ICAR-Institute of Wheat and Barley Research, Karnal. To match the booting stages of the two cultivars, staggered planting was performed. When 50% of the plants reached booting, heat stress was imposed for a period of eight days (5 °C more than the atmospheric temperature). The highest temperature to which the plants were exposed was 39 °C in the afternoon (1:00 to 2:00 p.m.). The plants were grown in a 16/8 photoperiod, adequate soil moisture was provided to the plants, and care was taken not to expose them to moisture deficit. They were irrigated once a day, preferably during the morning. The panicle tissues were collected in three biological replicates. Panicles of the same two cultivars grown in the adjacent field under ambient conditions were collected and used as ‘control’ samples. For expression analysis by qPCR, the field-grown plants were shifted to pots at the early tillering phase and kept in controlled growth chambers where an optimal temperature (32–35 °C) was maintained until booting. The temperature was increased to 38 °C upon booting, and the panicles were collected after eight days of heat stress so as to mimic the conditions used for the identification of the QTLs.

### 2.2. Datasets for the Identification of Heat-Responsive Genes in Rice

The microarray datasets of rice under heat stress carried out using the Affymetrix platform, available in the public domain (NCBI), were collected for meta-analysis and database development (Table 1). In total, there were seven GEO datasets from six different rice genotypes, including the data generated in our laboratory from N22 and IR64 panicles (GSE136746). The meta-analysis used 90 samples from seven datasets, which were represented by different tissues, namely the entire seedling (9), leaf and flag leaf (18), and panicle, including anther and pistil (59). The duration of heat stress treatment in these samples varied from 10 min to 8 days.

For generating the microarray dataset GSE136746, the total RNA was isolated (using Promega’s SV total RNA isolation system) from the panicles of two cultivars, N22 and IR64, grown under control and under 8 days of heat stress. The Affymetrix rice genome array (Affymetrix, Santa Clara, CA, USA) was hybridised with total RNA, and the microarray experiment was carried out according to the manufacturer’s procedure. The Affymetrix fluidics station 450 was used to wash and stain the probe arrays with anti-streptavidin-biotinylated antibodies. The Affymetrix^®^GeneChip^®^ Scanner 3000 was used for scanning the arrays.

### 2.3. Meta-Analysis and Database Construction

The normalisation and background correction of each dataset were performed using the robust microarray algorithm (RMA) in the Limma R package [44]. Further, the Combat tool was used to remove batch effects present in the datasets. Differentially expressed genes (DEGs) were identified under heat stress using the thresholds of average expression >12 and Log2 fold change (Log2FC) and were corrected for false discovery rate (FDR) with an adjusted *p* value of 0.05. For the correction of FDR, the Benjamini–Hochberg (BH) tool in the Limma package was used. The genes that satisfied these parameters were denoted as heat-responsive genes (HRGs). To this list of HRGs, the DEGs identified from the transcriptome were also added, and a database was constructed using the XAMPP (Apache, MariaDB, PHP, and Perl) server [45]. A MySQL relational database was used to design the backend of the database, while HTML5 and CSS3 were used for constructing the frontend. For constructing a user interface framework, JQuery and JavaScript were used [46,47,48]. For generating graphs, Chart.js was used [49]. PHP5 was used as a server-side scripting language to connect users and servers to access queries [45]. Following the databases on other biotic and abiotic stress-responsive genes developed earlier in the laboratory, this database was named RiceMetaSys-H, wherein ‘H’ stands for ‘heat or high-temperature stress’.

### 2.4. Construction of Protein–Protein Interaction (Epistasis) Network of Genes in the Known Major QTLs for Heat Stress Tolerance

All the genes located within the three major QTLs, one each on chromosomes 4, 5, and 9, were downloaded from the TIGR database (http://rice.tigr.org/, accessed on 25 May 2020), and their differential expression patterns in N22 and IR64 were analysed using the RiceMetaSys-H database developed. Only HRGs in the QTL regions were selected for further analysis. The STRING web portal (https://string-db.org, accessed on 7 November 2022) was used to identify the interacting partners of these QTL-HRGs. This was used as input in Cytoscape v.3.6.1, to identify their interaction network. For the selection of the candidate genes for gene expression analysis by qPCR, genes were selected based on their presence in the nodes of the major clusters.

### 2.5. Identification of Allelic Variants in the Key Heat Stress-Responsive Genes between IR64 and N22 in the Major Heat Stress QTL Regions

Using the whole genome sequence data generated in the laboratory for N22 and IR64 [50] and unpublished data, respectively, gene and protein sequences of all the candidate genes (QTL-specific genes and their interacting partners, as identified from the previous section) were aligned using the Os-Nipponbare-Reference-IRGSP-5.0 sequence as the reference. From this multiple sequence alignment, SNPs as well as amino acid changes (synonymous and non-synonymous) between the IR64 and N22 genotypes were identified. The genes in the QTL region harbouring non-synonymous substitution(s) between N22/IR64 were further subjected to expression profiling under heat stress.

### 2.6. Expression Analysis of the Identified Heat Stress-Responsive Genes

The panicles from N22 and IR64 grown under control and high-temperature conditions were collected and stored at −80 °C. RNA was isolated (Invitrogen, Thermo Fisher Scientific, Waltham, MA, USA) and converted to cDNA using the SuperScript^®^ III First-Strand Synthesis System and cDNA reverse transcription kit (Thermo Fisher Scientific). qRT-PCR was performed using an AriaMx real-time PCR system (Agilent, Santa Clara, CA, USA) machine and the KAPA SYBR FAST qPCR Master Mix (2X) reagent (Sigma-Aldrich, St. Louis, MO, USA). All qRT-PCR experiments were performed for the identified heat-responsive genes in both N22 and IR64 with three biological and technical replicates. The final reaction volume of 10 μL consisted of 50 ng cDNA, 0.5 μM of forward and reverse primer each, and 5 μL of KAPA SYBR FAST qPCR Master Mix (2X) reagent. The actin gene was used as a normaliser or internal control. The relative expression of the genes was calculated based on the Pfaffl formula (Ratio = 2-ΔΔCt) under control conditions and high-temperature conditions [51]. The details of all the primers used are provided in Appendix A.

## 3. Results

### 3.1. Part I: Identification of Heat Stress-Responsive Genes in N22 and IR64 under Heat Acclimatisation

The results of the microarray analysis of panicles of IR64 and N22 subjected to eight days of heat stress are briefly presented here. A set of 4731 DEGs were identified under heat stress, of which 950 were common to both genotypes, while 2924 and 2757 DEGs were specific to IR64 and N22, respectively (Figure 1A). Comparatively, more transcripts were up-regulated in IR64 (2674) than in N22 (1018) (Figure 1B). However, the degree of up-/down-regulation varied greatly in N22, compared to IR64. In N22, *DUF597* (Os01g0518000) was highly up-regulated by 127-fold, while the hsp20/alpha crystalline family gene (Os11g0244200) was 8.5-fold down-regulated. In IR64, the surface protein PspC (Os07g0599700) and the pectin esterase gene (Os05g0361500) were found to be 6-fold up-regulated and 5.3-fold down-regulated, respectively. The major accounts of the DEGs identified in both genotypes are briefly described in Figure 1C,D and Appendix A.

#### 3.1.1. Transcription Factors (TFs) and Genes of Transcriptional Regulation under Heat Stress

We found 247 (174 down and 73 up-regulated) and 212 (126 up and 86 down-regulated) DEGs encoding TFs in N22 and IR64 under heat stress (Appendix A). TFs such asAP2/EREBP, bHLH, bZIP, C2C2 Zn, C2H2 zinc finger family, CCAAT box binding factor family, homeo-box TFs, HSFs, MADS, MYB, TCP TF family, and WRKY TFs were the prominent up-regulated genes under heat stress. HSF genes are essential to sense the ROS signals and amplify the downstream genes in response to heat stress. Among HSFs, *OsHsfC2b* (Os06g0553100), *OsHsfC1b* (Os01g0733200), *OsHsfB2c* (Os09g0526600), *OsHsfA9* (Os03g0224700), and nucleic acid-binding, OB-fold-like domain-containing protein (Os03g0844900) were differentially regulated. Notably, *OsHsfC2b* (Os06g0553100) was 2.9-fold up-regulated even after 8 days of heat treatment in N22. In IR64, 10 HSF genes, *OsHsfC1b* (Os01g0733200), *OsHsfA2d* (Os03g0161900), *OsHsfB2a* (Os04g0568700), *OsHsfC2b* (Os06g0553100), *OsHSFB4b* (Os07g0640900), *OsHsfB2b* (Os08g0546800), *OsHsfB1* (Os09g0456800), *OsHsfB2c* (Os09g0526600), *OsHsfB2c* (Os09g0526600), and *OsHsfA2c* (Os10g0419300) were responsive to heat stress. Of these HSFs, only three, namely *OsHsfC2b* (Os06g0553100), *OsHsfC1b* (Os01g0733200), and *OsHSFB4b* (Os07g0640900), were down-regulated.

#### 3.1.2. HSPs and Other Chaperones

HSPs bind to proteins and stabilise intermediate stages of folding, assembly, degradation, and translocation across membranes. Among the identified 32 heat stress-responsive HSPs and other chaperones in N22, one HSP70 encoding gene (Os11g0187500) and three genes of DNAj heat shock N-terminal domain-containing proteins (Os09g0493800, Os05g0374500, and Os02g0782300) were up-regulated even after 8 days of heat stress. However, none of the sHSPs were up-regulated in the panicles of N22 after 8 days of heat stress. In the case of IR64, a total of 35 genes encoding HSPs and other chaperones were found to be heat stress responsive. Among them, five out of nine DNAj proteins-encoding genes, five HSP70 genes, one HSP90, one HSP100, seven small HSP genes, and one endoplasmic reticulum chaperone (*OsBiP1*/Os02g0115900) were up-regulated (Appendix A).

#### 3.1.3. Hormone Metabolism and Signal Transduction-Related Genes

A total of 52 genes related to auxin, cytokinin, gibberellin, ABA, ethylene, JA, SA, and BA metabolism, as well as signal transduction, were found to be heat stress-responsive in N22 panicles, while in IR64, 27 genes of auxin metabolism, six of ABA metabolism, two of brassinosteriods, 22 genes of ethylene metabolism, four genes of cytokinin metabolism, three genes of jasmonate metabolism, five genes of gibberellin metabolism, and one gene of salicylic acid were found to be up-regulated under heat stress (Appendix A). Calcium signalling-related genes such asauto-inhibited calcium ATPases (Os08g0517200 and Os02g0176700), calcineurin B-like proteins (Os12g0162400 and Os12g0597000), a calmodulin-binding region domain-containing protein (Os01g0708700), and an EF-hand domain-containing protein (Os02g0158100) were up-regulated in N22. Five calmodulin-binding region domain0containing proteins, four calmodulin genes, three calreticulin, four EF-Hand-type domain-containing proteins, and one each of calnexin, calcineurin B-like protein9, and endoplasmic reticulum-type Ca2+-ATPase1 were among the up-regulated genes of calcium signalling in the IR64 panicle.

#### 3.1.4. ROS-Related Genes

There were 33 ROS-related HRGs in both N22 and IR64 panicles in this study (Appendix A). One of the superoxide dismutase genes, Os06g0115400, maintained higher expression even after 8 days of heat stress in N22. All 12 thioredoxin-related genes, except Os06g0163400, were repressed upon heat stress in N22. Similarly, all peroxiredoxin and glutaredoxin-related genes were also down-regulated upon stress in N22. On the contrary, all 10 thioredoxin genes except three (Os04g0629500, Os07g0684100, and Os02g0774100) were up-regulated in IR64. The antioxidant enzymes such as catalase (Os02g0115700) and superoxide dismutase (Os03g0351500) were also found to be significantly up-regulated. Antioxidants such as ascorbate- and glutathione-related genes such as cytochrome family genes (Os01g0971500; Os02g0642300; Os05g0108800; Os10g0504200), ascorbate free radical reductase (Os09g0567300), L-ascorbate peroxidase (Os03g0285700; Os07g0694700), thylakoid-bound ascorbate peroxidase (Os04g0434800), glutathione peroxidase (Os04g0556300), and GDP-L-galactosephosphorylase (Os12g0190000) were also up-regulated significantly in IR64. Thus, it is clear that the heat acclimation of the more tolerant genotype, N22, is not primarily due to the redox mechanism.

#### 3.1.5. Cell Division and Cell Cycle

Ten and eleven genes involved in cell division and the cell cycle, respectively, were found to be responsive to heat stress in the heat-tolerant rice genotype N22 (Appendix A). Cell-division-related genes such as mitotic checkpoint family protein (Os01g0877300), regulator of chromosome condensation (RCC1) family protein (Os05g0456925), SMC5protein (Os05g0596600), CDC45-like family proteins (Os11g0128400; Os12g0124700), and septum formation topological specificity factor MinE family gene (Os12g0498400) and some cyclin-related genes were up-regulated. Five out of six genes related to cell division, such as UVB-resistance protein (Os02g0554100), regulator of chromosome condensation (Os03g0599600; Os05g0456925), mitotic spindle checkpoint protein MAD2 (Os04g0486500), and CDC45-like protein family protein (Os12g0124700) were up-regulated significantly in IR64 panicles upon heat stress. The cell division inhibitor gene (Os02g0834700) was down-regulated upon heat stress in IR64. Further, 20 out of 24 genes involved in the cell cycle were up-regulated in IR64.

#### 3.1.6. Ubiquitin-Dependent Degradation

Forty-five genes related to the ubiquitin-mediated protein degradation pathway were found to be responsive to heat stress in N22. Of these, 19 (42%) genes such as F-box family protein-coding genes (Os08g0461100; Os08g0193900; Os08g0461000), Cullin1 (Os01g0369200; Os04g0484800), cell-death-related proteins (Os03g0275900; Os03g0275900), ubiquitin-activating enzyme (E1) (Os02g0506500), and ubiquitin-conjugating enzyme E2 (Os04g0580400) were up-regulated. However, in response to heat stress, ubiquitin ligase (E3) was down-regulated. In heat-stressed IR64 panicles, 50 genes belonging to this pathway were differentially regulated. The majority of these genes (80%) were up-regulated; importantly, plant U-box-containing protein 27 (*OsPUB27*/Os04g0489800) was up-regulated approximately8.3-fold (Appendix A).

#### 3.1.7. Sucrose-Starch Pathways

Heat stress activated nine and fifteen genes related to sucrose and starch metabolism, respectively, in N22 panicles (Figure 1E; Appendix A). Sucrose transporter (Os02g0576600), vacuolar invertase (Os04g0535600), cell wall invertase (Os09g0255000), and sucrose phosphatase (Os05g0144900) were slightly down-regulated approximately2–3-fold, whereas hexokinase (Os01g0742500; Os01g0940100; Os05g0532600) and fructokinase genes were up-regulated upon stress. Starch biosynthetic rate-limiting enzyme AGPase (Os05g0580000; Os03g0735000) was up-regulated. Among the differentially regulated starch synthases (Os05g0533600; Os02g0744700) and granule-bound starch synthases (Os06g0133000; Os07g0412100), all were up-regulated except Os02g0744700. The important enzyme of the starch biosynthetic pathway, a starch-branching enzyme (Os06g0726400), was slightly (2.2-fold) up-regulated. Interestingly, starch-cleaving enzymes such asα-amylase (Os08g0473900) and β-amylase (Os03g0141200; Os10g0465700), were down-regulated. Of these, α-amylase was highly down-regulated approximately14-fold. The plastidial disproportioning enzyme (Os07g0627000) and maltose excess protein 1 chloroplast precursor (Os04g0602400) were also slightly up-regulated.

There were 12 DEGs related to sucrose biosynthesis and degradation in IR64 under heat stress (Figure 1F). The sucrose synthase (Os03g0401300) gene was 28-fold up-regulated in the heat-stressed panicles of IR64, followed by cell wall invertase (Os02g0534400; Os04g0413500), neutral invertase, sucrose-phosphate synthase, sucrose phosphatase and hexokinase. This showed that the entire pathway, from sucrose synthesis to transport, followed by their breakdown into the constituent monosaccharides, was up-regulated. Nine genes related to starch biosynthesis were also up-regulated in IR64, including AGPase, starch synthase, granule-bound starch synthase, α-amylase, β-amylase, and cytosolic starch phosphorylase. Granule-bound starch synthase (Os07g0412100) and the large subunit of AGPase were also up-regulated in IR64 panicles upon heat stress, similar to N22. Though the starch synthesis was up-regulated in IR64 under heat stress, like that of N22, the starch was also simultaneously broken down in IR64. When coupled with the up-regulated sucrose transport and starch catabolism, the heat-modulated sucrose-starch pathway resulted in poor grain filling in IR64 [25].

#### 3.1.8. Expression Pattern of Microspore/Pollen and Tapetum-Specific Genes

Suwabe et al. [52] analysed the transcriptome of developing microspore and tapetum tissue to identify genes involved in the development of pollen in rice. With reference to these data, we found 12 pollen-specific, 2 tapetum-specific, and 6 genes expressed in both tissues to be heat stress-responsive in N22 panicles. Only two out of 12 pollen-specific genes (ubiquinone oxido-reductase-LOC_Os01g52214 and ubiquitin-specific protease-Os01g0771400) were up-regulated and the remaining genes were down-regulated. Pollen-specific genes that were found to be down-regulated in this study included L-ascorbate oxidase (Os01g0816700), bacterial transferase hexapeptide repeat domain-containing protein (Os01g0919200), ripening-associated protein (Os03g0796000), pectin methyl esterase inhibitor (Os07g0247000), β-amylase (Os10g0465700), beta-expansin (Os10g0548600), and pollen allergen (Os04g0317800) (Appendix A). The tapetum-specific acyl carrier protein III (Os09g0539800) was up-regulated under heat stress. Two pollen allergen genes (Os03g0808500and Os04g0465600) that have been reported to be pollen- and tapetum-specific were found to be up-regulated in N22.

In IR64, 22 genes of microspore- or pollen-specific genes were found to be heat stress-responsive, all of which were up-regulated. Some of them included DNA methyl transferase (Os03g0226800), L-ascorbate oxidase (Os01g0816700), actin (Os03g0718100), pollen allergen (Os04g0317800; Os06g0655200; Os03g0808500; Os04g0465600), pectin methyl esterase inhibitor (Os07g0247000), PRLI-interacting factor F (Os10g0406100), β-amylase (Os10g0465700), β-expansin (Os10g0548600), bacterial transferasehexapeptide repeat domain-containing protein (Os01g0919200), acyl carrier protein III (Os09g0539800), BRITTLE CULM-LIKE 4 (Os05g0386800; 5-fold), and phytochelatinsynthetase (Os05g0386800).

### 3.2. Part II: Meta-Analysis and Database Creation for Heat Stress-Responsive Genes in Rice

For the meta-analysis of heat stress-responsive genes, seven rice microarray datasets, including the in-house dataset (GSE136746) described in the previous section, were chosen. This dataset represented six genotypes, of which two were tolerant (N22 and Cultivar 996) to heat stress while the other four were sensitive (Moroberekan, M202, IR64, and Pusa Basmati) to heat stress (Table 1). The duration of heat stress varied from 10 min to 8 days in these datasets. After the removal of batch effects, normalisation, and differential expression analysis, 4227unique HRGs were identified (adjusted *p*-value of 0.05; Log2 (fold change value) of <−1 to >1, and average expression value >12). For arriving at these results, HRGs were identified using different parameters, namely, Log2 fold change (FC) values of 1, 2, and 3 as well as average expression values of 10, 12, and 14 (Appendix A), from which the most optimal dataset was chosen. The scheme of data analysis and the architecture of the database are given in Figure 2. The construction of the database is explained in detail in the methods section.

The features of the database have been designed to cater to the needs of crop improvement researchers, akin to its predecessor databases, RiceMetaSysA and RiceMetaSysB [41,43]. The prominent feature of the HRG database included searching for the HRGs through genotypes, growth stages, tissues, physical intervals in the genome (to enable search in the QTL intervals), and Locus IDs, all of which provide complete information on the HRGs with their annotation, fold change, the direction of gene regulation, and the experimental material used for the analysis. An option has been included to find HRGs that are common to two or more genotypes using the ‘common among varieties’ tool, which gives an idea about the nature of the HRGs in multiple genotypes. A provision has been made to retrieve the DEGs among the heat tolerant and sensitive genotypes using the search option ‘tolerant vs. sensitive’. This provision is expected to be particularly useful for making decisions on whether to include the HRG in genomic selection or not. SSR markers in the selected HRGs can be obtained using the SSR tab to enable the marker-assisted selection of the candidate genes (Appendix A). Additionally, an external link to ‘PlantGSEA’ is provided to help in the enrichment of datasets to identify robust candidate genes. Graphical display has been enabled in the HRG database to compare gene expression levels of either up to 10 HRGs within a genotype or a single HRG across all the genotypes for a given gene (Locus ID). When a large number of genes are to be compared, then the external link to HeatMapper can be used.

#### HRGs Identified across Developmental Stages, Tissues, and Their Pathway Analysis

Meta-analysis identified a large number of HRGs from rice panicles (2678) followed by flag leaves (1424). Both the reproductive tissues, pistil and anther, had the lowest number of HRGs, 56 and 167, respectively (Figure 3A). The number of up- and down-regulated genes were nearly equal in all tissues except panicles, where the number of down-regulated genes (1656) was much higher than that of up-regulated genes (972), a pattern that was also observed in N22 panicles in the dataset, GSE136746. Among the developmental stages, the reproductive stage was found to be more sensitive to heat stress, with 79.54% (3360 out of 4227) of the genes being heat stress responsive, while only one-fourth of the total HRGs were heat stress responsive during early developmental stages (Figure 3B). Among the seven genotypes, a heat stress-tolerant genotype, cultivar 996, was found to have a disproportionately large number of HRGs (3959) compared to the rest of the genotypes studied, except Pusa Basmati with 1119 HRGs (Figure 3C). In a heat stress-sensitive genotype, M202, no HRGs could be identified from meta-analysis after FDR correction. Among the 4227 unique HRGs (after removing the duplicates that were common to two or more genotypes), 39% belonged to cellular components, 35% to biological processes, and the remaining 26% to molecular responses (Figure 3D).

A substantial proportion of the HRGs showed down-regulation in the heat-sensitive genotypes Moroberekan (20/31) and IR64 (12/21) (Appendix A). Hsp70, Hsp82, and many carbon-metabolism-related genes such as isocitrate dehydrogenase, phosphoglycerate kinase, and glyceraldehyde 3-phosphate dehydrogenase were all down-regulated, as were ROS pathway genes such as L-ascorbate peroxidase and glutathione reductase in IR64, and ADP malate, vacuolar ATP, and auxin biosynthesis pathway genes in Moroberekan (Appendix A). The up-regulated ones in Moroberekan included only a few genes, such as those involved in calcium-related metabolism (e.g., calnexin and calreticulin) and some carbon fixation genes (e.g., chlorophyll a- and b-binding proteins). Similarly, in the case of IR64, only the smaller subunit of RuBP carboxylase, PSII oxygen-evolving complex genes, and Hsp70 were up-regulated. The hydroxyl pyruvate reductase-encoding gene, one of the major photorespiration-related genes, which reduces the carbon fixing ability, was up-regulated in IR64 under heat stress.

The pathway analysis revealed that enzymes involved in carbon fixation and the ROS pathway were up-regulated in both the heat stress-tolerant genotypes, N22 and cultivar 996 (Appendix A). In N22, pectate lyases and Ca^++^-regulating genes were down-regulated, while pectin lyases, which do not have an absolute requirement of Ca ions for their action, were up-regulated. In cultivar 996, besides better starch and sucrose metabolism, the biosynthesis and signalling genes of phytohormones, cytokinin and ethylene, were up-regulated, while those of ABA and brassinosteroid were down-regulated. While N22 showed the up-regulation of photosynthesis-related genes, cultivar 996 did not (Appendix A). Further, chloroplast and porphyrin biosynthesis genes were down-regulated under heat stress in cultivar 996. Among the Hsps, many high-molecular-weight Hsps such as Hsp 70, Hsp 90, and Hsp 81.3 were up-regulated in cultivar 996, while chloroplast Hsp70 was down-regulated. HSP 17.3 was up-regulated among the low-molecular-weight HSPs, whereas HSP 20 was down-regulated. In addition to HSP 70, many low-molecular-weight HSPs such as HSP 17.5, HSP 17.9, and HSP 18 were up-regulated in Moroberekan.

In Pusa Basmati, the output of the pathway analysis included only 96 HRGs, of which one half was up-regulated and the other half was down-regulated. The major up-regulated genes included DNAJ Hsp, HSP 70, 60S ribosomal protein genes, and pyruvate and carbon metabolism-related genes such aspyruvate kinase and fructose bisphosphatase. The down-regulated genes included phosphofructokinase, 40 and 50S ribosomal genes, granule-bound starch synthase Ib, RCCR1, and oxidative photophosphorylation-associated genes (Appendix A). Plant hormone regulation was not appreciably modulated under heat stress in this genotype.

### 3.3. Part III: Utilisation and Validation of the Database following the Identification of HRGs from QTL Regions and Their Interacting Partners

#### 3.3.1. Selection of Genes and Genomic Regions for Validation

The major effect QTLs identified from the N22/IR64 mapping population were considered for database use and subsequent validation. This included the two major effect QTLs reported by Shanmugavadivel et al. [25] on chromosomes 5 (*qSSIY5.2*) and 9 (*qSTIPSS9.1*) and another major QTL on chromosome 4 (*qspf4.1*) identified using the same population and genotype dataset but at a different location (National Rice Research Institute, Cuttack, India; unpublished results).All the details of the three QTLs are given in Appendix A. There were a total of 134 genes, i.e., 18, 54, and 62 genes, in the QTL intervals on chromosomes 4, 5, and 9, respectively (Appendix A).

These 134 genes were looked up in the database, RiceMetaSys-H, for their responsiveness to heat stress. The genes that showed differential expression in the database were used as input in the STRING database to identify the individual gene networks (unweighted). The retrieved networks were merged, and their cluster analysis was performed. This analysis identified a total of 16 hub genes (from the top two major clusters) centred on chromosomes 4 and 9 as the key regulators of the heat stress response (Figure 4). These 16 hub genes interacted strongly with 41 other genes, which are the hub genes’ close interactive partners. The majority of these genes were associated with either HSFs or sugar metabolism. The entire set of hub genes and their close interactive partners, summing up to 57 candidate genes (Appendix A) distributed on chromosomes 3, 4, 5, 8, and 9, which harboured 4, 12, 14, 8, and 19 genes, respectively, were selected for further validation by expression profiling.

#### 3.3.2. Identification of Allelic Variants in the Key Heat Stress-Responsive Genes between IR64 and N22 in the Major Heat Stress QTL Regions and Network Genes

The allelic variant analysis used Nipponbare as the reference. Among the 134 genes, 55 genes were annotated as either transposons or retroposons or expressed or hypothetical proteins. Of the remaining 79 genes, 47 genes showed SNPs only in the intergenic regions. Only in the rest of the 32 QTL-specific genes could we identify SNPs in the genic regions. The number of SNPs in IR64 varied from 1 to 35, and each of the 32 genes had at least one SNP. In the case of N22, six genes did not harbour any SNPs, and the number of SNPs varied from one to thirty-two per gene. Overall, N22 had fewer SNPs (217) as compared to IR64 (352) (Figure 5A; Appendix A). The same trend was seen in the number of transitions and transversions, as well. In N22, there were 6, 8, and 14 genes that showed no SNP changes, transitions, and transversions, respectively, and the genomic sequences were almost similar to the reference in most of the genes except LOC_Os05g43810 and LOC_Os09g27590, which had 12 transversions each (Appendix A). Both of these genes were expressed proteins. In contrast, in IR64, there were only five genes that had no transversions. More importantly, none of the SNPs in these 32 genes were similar between IR64 and N22, clearly suggesting that the variations identified were distinct mutations, specific to these genotypes (Appendix A).

The amino acid sequence alignment showed that three genes in IR64 and ten genes in N22 had no changes with respect to Nipponbare. However, these latter 10 genes had amino acid substitutions in IR64 (Appendix A). Overall, twenty-five genes showed distinct amino acid changes in IR64 and N22, while only six showed identical changes in both genotypes. A few genes having distinct changes in N22 and IR64 are the myosin heavy chain-related gene (LOC_Os05g43870), *OsSigP6*—putative Type I Signal Peptidase homologue that employs a putative Ser/Lys catalytic dyad (LOC_Os09g28000), and phosphate carrier protein and putative mitochondrial precursor (LOC_Os09g28160). Overall, there were 80 unique and 27 common amino acid substitutions between the N22 and IR64 genotypes, with respect to Nipponbare.

The 57 network genes showed a contrasting scenario of more SNP changes in N22 as compared to IR64 (Appendix A and Figure 5B). While IR64 harboured 429 SNPs, 274 transitions, and 155 transversions, N22 had comparatively more changes (473, 299, and 174, respectively). On a closer look, only a few genes harboured a higher number of changes (>20 per gene) in the two genotypes (Figure 5B).The maximum number of transversions were harboured in the same gene, LOC_Os08g31980, in both IR64 (16) and N22 (18) (Appendix A). However, IR64 and N22 differed by only two transversions in LOC_Os08g31980, with all the other SNPs being identical. This trend was seen in almost all the network genes. Out of the 57 network genes, 24 genes had identical amino acid sequences in IR64 and N22 as well as the reference genome Nipponbare. Eight of the remaining thirty-three genes showed similar amino acid changes between IR64/N22. Thus, only 25 genes had amino acid changes between N22/IR64, which included 7 genes with no changes in IR64 and 4 genes with no changes in N22. Of the remaining 14 genes, UDP-glucuronosyl and UDP-glucosyltransferase domain-containing protein (LOC_Os09g34250), phosphate carrier protein and mitochondrial precursor (LOC_Os09g28160), calreticulin precursor protein (LOC_Os04g32950), and heat shock protein (LOC_Os04g01740) showed a higher number of non-synonymous substitutions. Overall, the amino acid variations were comparatively fewer in the 57 network genes, with just 50 distinct and 38 common amino acid substitutions between N22/IR64.

#### 3.3.3. Expression Analysis of the Heat Stress-Responsive Genes Identified

Between the 32 genes from the QTL region and the 57 genes from the network analysis, 8were found to be common (LOC_Os04g42020, LOC_Os05g44100, LOC_Os09g27610, LOC_Os09g27650, LOC_Os09g27830, LOC_Os09g28050, LOC_Os09g28160, and LOC_Os09g28200). A total of 89 genes were considered for expression analysis under two treatments: control and 8 days of heat stress in N22 and IR64 (Appendix A). Primer details for the 89 genes selected for expression profiling under heat stress are given in Appendix A. Out of the 57 network genes, 27 were differentially expressed between IR64 and N22, while the remaining30 showed a similar expression pattern in both genotypes, i.e., 20 and 10 of these 30 were up- and down-regulated, respectively, in both IR64 and N22 (Appendix A). The maximum number of DEGs were from chromosome 9 (11 DEGs), while none of the genes from chromosome 3 showed any differential expression pattern. The DEGs from chromosome 9 were related to sugar/protein biosynthesis, HSFs, cell structure related-protein coding genes such as galactosyltransferase, putative, expressed (LOC_Os09g27950), UDP-glucuronosyl and UDP-glucosyltransferase domain-containing protein (LOC_Os09g34250), aspartate aminotransferase (LOC_Os09g28050), HSF-type DNA-binding domain-containing protein (LOC_Os09g35790), microtubule associated-protein (LOC_Os09g27700), and annexin (LOC_Os09g27990). There were five more DEGs from chromosomes 4, 5, and 8: glycosyl hydrolases family 16, (LOC_Os04g51460), glycosyl hydrolase (LOC_Os05g15880), trehalose synthase (LOC_Os05g44100), trehalose-6-phosphate synthase (LOC_Os05g44210), and trehalose-6-phosphate synthase, putative, expressed (LOC_Os08g31980).

There were a total of 32 genes for qRTPCR, including 5, 15, and 12 from chromosomes 4, 5, and 9, respectively, in the QTL-specific genes. All 32 genes tested were heat stress-responsive. Seventeen of the thirty-two genes showed a differential expression pattern between the two contrasting genotypes, while of the rest of the fifteen genes, seven were up-regulated and eight were down-regulated in both genotypes (Figure 6). The distribution of the 17 DEGs showed that 3, 7, and 7 DEGs were from chromosomes 4, 5, and 9, respectively (Figure 6). The DEGs were found to be related to cell structure integrity and carbohydrate metabolism (LOC_Os05g43870), myosin heavy chain (LOC_Os05g44100), trehalose synthase (LOC_Os09g28150), bifunctional monodehydroascorbate reductase, and carbonic anhydrase nectarin-3 precursor.

## 4. Discussion

Each abiotic stress is unique in its manifestation and management. While nutritional stress, including water-deficit stress, can be ameliorated externally by supplementation, salt stress persists throughout the life cycle of the crop, and amendments are made to the soil during the fallow period. Heat stress takes a middle path, wherein the higher temperature persists for a specific period of time but not the entire crop period and cannot be ameliorated externally. Thus, for combating heat stress episodes and maintaining crop yield and quality, genetic improvement alone remains the most effective strategy.

Most of the studies on genome-wide transcriptome dynamics invariably focus on a short period of heat stress, while QTL studies better mimic the field scenario and impart a longer duration of heat stress (Table 1) [18,25]. The longest duration of heat stress imposed so far, in the transcriptome datasets generated to date, is three days of heat stress (Table 1). Because the reproductive phase of rice is the most sensitive to heat stress, we imposed heat stress at the start of this phase, as in previous studies [12,38], but we extended it for eight days, unlike them, to mimic the pattern of heat incidences in field conditions. The genotypes selected for the present study were the parents of the mapping population used to identify two major QTLs, one of which is a novel one [25]. Using a genome-wide transcriptional dynamic study, we targeted the reproductive sink tissue and a longer period of heat stress to identify heat stress acclimatisation genes and pathways in rice. Here, we have used only the microarray data of heat stress tolerance as, in 2021, a comprehensive analysis of RNA-seq data under heat stress tolerance has been published with an emphasis on alternative splicing [53]. Further, in RNA-seq data, one needs to conduct the analysis experiment by experiment [53] which makes meta-analysis not possible, at least to our knowledge. Most of the low-abundance transcripts also need to be removed for normalisation and this takes away the advantage of using the RNA-seq data.

Starch synthesis in higher plants is catalysed by AGPase, GBSS, SS, SBE, and DBE, with AGPase being responsible for the first rate-limiting step [54,55,56]. Most starch biosynthetic pathway genes, such as *AGPase*, starch synthase, and starch-branching enzymes, were found to be up-regulated in heat-stressed N22 panicles, which may explain why N22 has the highest fertility at higher temperatures. In IR64 also, most of the starch synthase isoforms were up-regulated, including the AGPase large subunit coding gene, though the rest of the starch biosynthetic genes remained unaltered. The α- and β-amylases are key enzymes in the conversion of starch to dextrin and maltose. α-amylase was significantly down-regulated (14-fold) in N22 panicles but up-regulated in IR64. This indicates that the hydrolysis of stored starch was not carried out under heat stress in N22, whereas stored starch was broken down, leading to reduced spikelet fertility under heat stress in IR64. Shi et al. showed recently that soluble starch concentration levels significantly increased in N22 after 6 days of heat stress (both at 38 °C and 40 °C) at the heading stage as compared to a control. They also showed that in IR64, the starch concentration decreased steeply by 23% at 40 °C and by 5% at 38 °C, in comparison with the control. Thus, our observation and proposed mechanism of heat stress tolerance have a better footing with the observations made by Shi et al. [57]. The amylose content of rice endosperm is regulated at the level of Waxy gene transcript processing at the stage of intron I excision from the Waxy pre-mRNA [58], and this gene transcript was also found to be heat stress-responsive and up-regulated in IR64.

*OSINV4* (Os04g0413200) encodes a cell wall invertase that is expressed in the whole panicle, particularly in the anthers, but is repressed by cold stress. In cold stress-tolerant rice cultivars, however, its expression was found to be unaltered under cold stress [59]. We also found its expression to be unaltered under heat stress in N22 panicles. This indicates that cell wall invertase works well in N22 under heat stress to maintain starch levels in pollen grains, leading to better pollen fertility of N22. Further, cell wall modification genes have a major role in the development of pollen and rice grains, and they are different from those of Arabidopsis [60]. Lin et al. [61] reported a positive correlation between levels of sHSP in caryopsis and the chalky trait, as they found elevated levels of sHSP in the caryopsis of chalky rice genotypes. sHSP20 has been demonstrated to impart enhanced heat and salinity tolerance in rice and other organisms [62]. We found the down-regulation of sHSP expression in N22, which might correlate with the non-chalkiness of their grain under heat stress. In contrast, sHSP expression was up-regulated in IR64, which might correlate with the chalkiness of IR64. The de-esterification process by pectin methyl esterase (PME) is known to promote pollen tube growth. The pollen-specific PME inhibitor (Os07g0247000) was found to be highly up-regulated (7-fold) in the heat-susceptible genotype IR64 panicles in the current study. That could explain the improper development of pollen tube growth and subsequent sterility in IR64 under heat stress.

We created a database based on our previous work on other abiotic and biotic stress tolerances [41,43] in order to broaden our search for expression dynamics under heat stress. Heat stress had an impact on a large number of genes, as high as 11,229, as identified by meta-analysis (Appendix A). To make the results more reliable, a high average expression of 12 and Log2 FC1 was adopted, which downsized it to 4227 genes. Our database has the inventory of the HRGs identified and provides multiple search options, for use by the research community, particularly, for the rice breeders involved in the improvement of heat stress tolerance. Rice stress-responsive genes common to the seedling stage were retrieved from the RiceMetaSysA (for drought and salinity-responsive genes) and the current database, and the major hub genes of multiple stress tolerance in rice were identified and validated [63]. We also identified genes that are common to all three abiotic stresses in rice across all tissues and developmental stages (Figure 7 and Appendix A). Overall, we found 349 genes to be common across the three stresses. This could serve as a useful resource for allele mining to identify superior alleles that could in turn be used in developing climate-resilient varieties. Rapid evolution in sequencing technology and genomic resources has shortened the long and tedious process of genetic map-based gene discovery by contributing in myriad ways. We anticipate using the information provided by the database to dissect QTLs, in particular. So, we have tried to integrate the information on gene expression developed in the present study and the whole genomic (resequencing) resources to narrow down the candidate genes for heat stress tolerance in the QTL regions of yield and spikelet sterility, qSTIY5.1 and qSTIPSS9.1.

The observations that (i) the number of transitions and transversions in the two genotypes were nearly equal in the case of the network/epistatic genes with respect to the reference genome, (ii) the number of SNPs was nearly 33% higher in IR64 than N22 in QTL genes, and, (iii) more importantly, the SNPs were nearly identical for the network genes while they were more distinct between N22 and IR64 for the QTL-specific genes, added support to the identity of the QTL regions. As a result, we propose that QTL regions can be directly transferred by marker-assisted breeding to improve recipient cultivars’ heat stress tolerance.

The amino acid substitution frequency was found to be 0.856/Kbp and 0.289/Kbp for N22/IR64-specific (distinct) and common variants in the QTL region. In the case of network genes, the frequency was nearly equal at 0.274/Kbp and 0.208/Kbp for the distinct and common variants, respectively. In other words, the proportion of unique amino acid substitutions (between N22/IR64) in the QTL-specific genes was much higher than the common substitutions, i.e., 2.58:0.88 (2.93-fold), compared to the network genes at 0.88:0.67 (1.313-fold) ratio. Our observation clearly demonstrated that the directional selection for abiotic stress tolerance could be observed only in the causal genes, relevant variations accumulated in the niche regions (N22 was bred from landraces of abiotic stress-prone places), while the other genic regions accumulated variations randomly without undergoing any selection pressure across genotypes.

Expression profiling under heat stress lent support to five of the twenty-seven genes in the QTLs *qSTIY5.1* and *qSTIPSS9.1*, namely LOC_Os09g27650, LOC_Os05g44090, LOC_Os05g44320, LOC_Os05g43850, and LOC_Os05g44190, while the HRG database lent support to four of these genes (all except LOC_Os05g44090) besides LOC_Os05g43870, LOC_Os09g27650, and LOC_Os09g28000. Thus, in the true sense, the two major QTLs identified were QTLs with multiple genes contributing to the trait expression. In the third QTL, *qspf4.1* (unpublished), expression analysis lent support to a single gene, LOC_Os04g444140, annotated as non-symbiotic haemoglobin 2, while in the HRG database, we could not find any differentially expressed genes in this region. The genes identified in the present study can be used through genome editing for climate-resilient rice development.

To extend this work to other crops, for the four HRGs that were common to the QTL study, expression analysis, and HRG database, we identified the orthologues from Arabidopsis, wheat, maize, and *Setaria italica*. BlastP results showed that the Poaceae (grass) orthologues were closer to the auxin response factor gene, LOC_Os05g43870, and protein-disulphide isomerase 2-3-like gene, LOC_Os09g27830, of rice, while the Arabidopsis orthologue was closer to C2H2-like zinc finger-like protein, LOC_Os09g27650. LOC_Os09g28000 encoding for thylakoid processing peptidase1 shared better homology with both the dicot and monocot members (Appendix A).

Thus, our work has contributed a database for heat stress-responsive genes in rice and identified the nine most promising genes for further functional validation as well as breeding use in combating climate change in general and high-temperature stress in particular.

## Figures and Tables

**Figure 1 plants-12-01697-f001:**
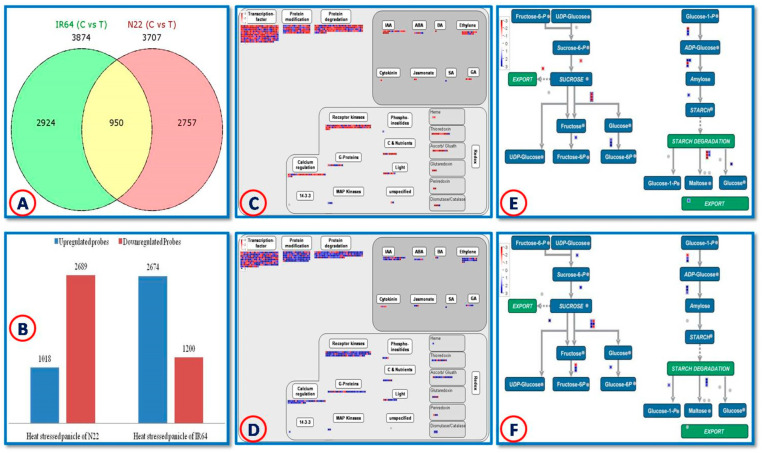
Microarrayresults of differentially expressed genes (DEGs) in the panicles of IR64 and N22 under eight days of heat stress. (**A**): Number of genotype-specific and common DEGs in IR64 and N22 in response to heat stress; (**B**): number of up-regulated and down-regulated genes in IR64 and N22 in response to heat stress; (**C**): MapMan overview of the major pathways of the DEGs of heat-stressed N22 panicles; (**D**): MapMan overview of the major pathways of the DEGs of heat-stressed IR64 panicles; (**E**): MapMan overview of sugar–starch pathway DEGs in the heat-stressed N22 panicles; (**F**): MapMan overview of sugar–starch pathway DEGs in the heat-stressed IR64 panicles.

**Figure 2 plants-12-01697-f002:**
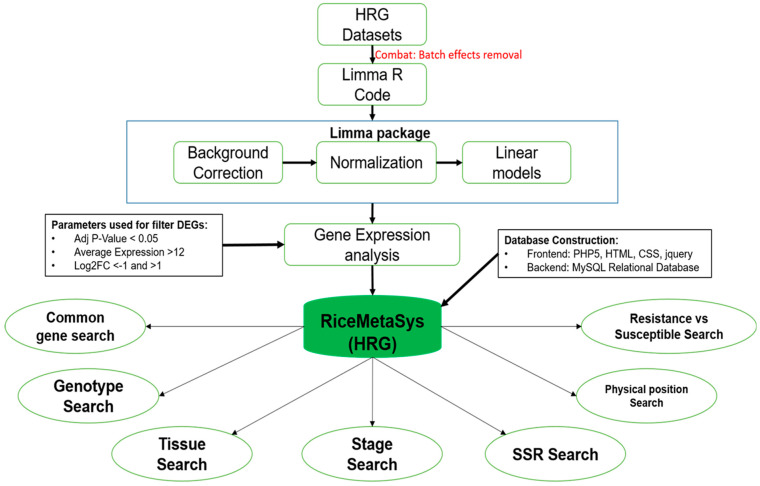
Schematic representation of data analysis and construction of the RiceMetaSys-H database. It shows the meta-analysis pipeline followed for microarray data analysis and the search provisions enabled in the database. The tools used and the statistical parameters are also shown.

**Figure 3 plants-12-01697-f003:**
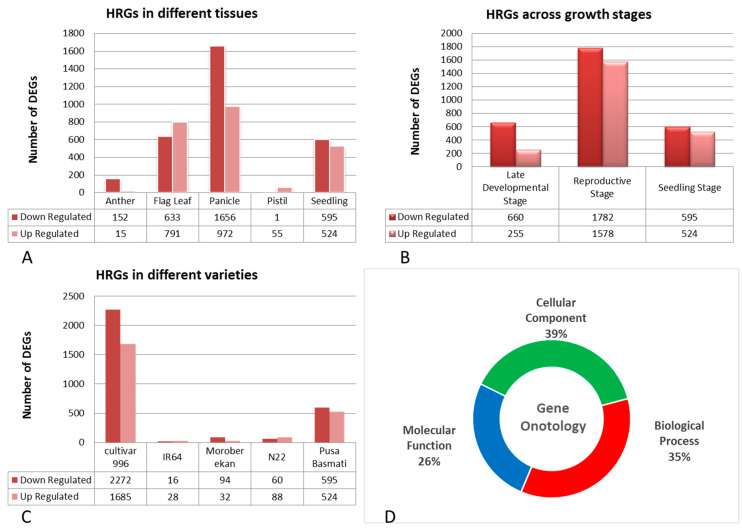
Meta-analysis of rice microarray datasets across different stages and tissues under heat stress. (**A**) Tissue-specific heat stress-responsive genes; (**B**) developmental-stage-specific heat stress-responsive genes; (**C**) genotype-specific heat stress-responsive genes; (**D**) Gene Ontology of the differentially expressed genes (DEGs).

**Figure 4 plants-12-01697-f004:**
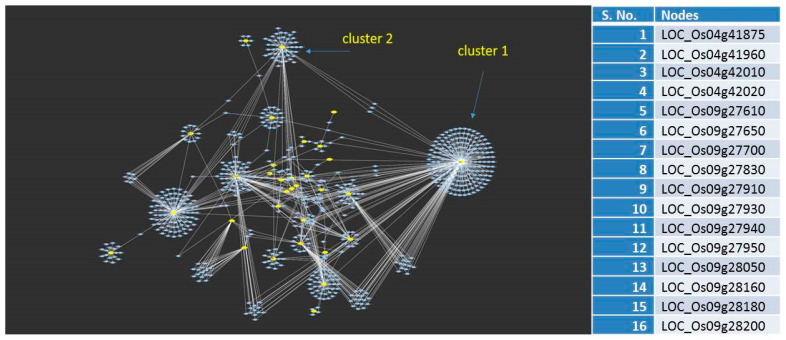
Network analysis of the candidate genes present in the selected QTL regions on chromosomes 4, 5, and 9 identified using recombinant inbred lines developed from N22/IR64. The 16 nodes obtained from the network analysis are presented in the right panel.

**Figure 5 plants-12-01697-f005:**
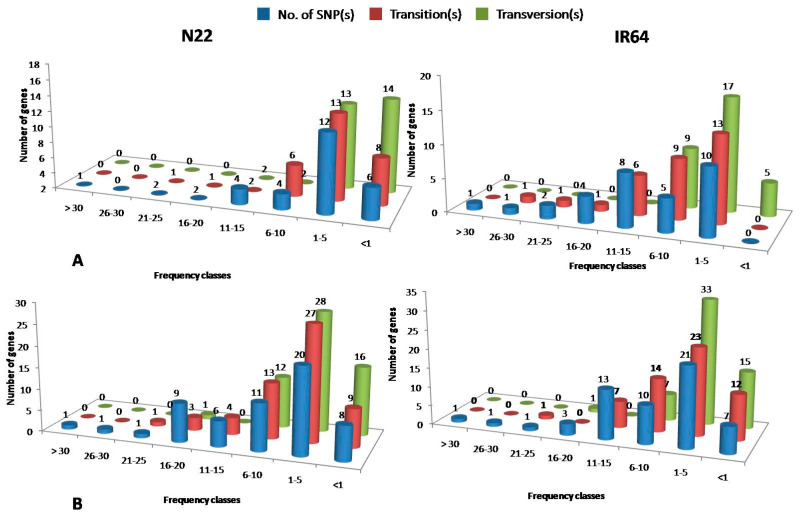
Frequency distribution of the genes showing SNP including transitions and transversions, observed in N22 and IR64 in the (**A**) selected QTLs for heat stress tolerance identified on chromosomes 4, 5, and 9 using recombinant inbred lines developed from N22/IR64, and (**B**) for the major hub genes identified through network analysis of the QTL-specific genes.

**Figure 6 plants-12-01697-f006:**
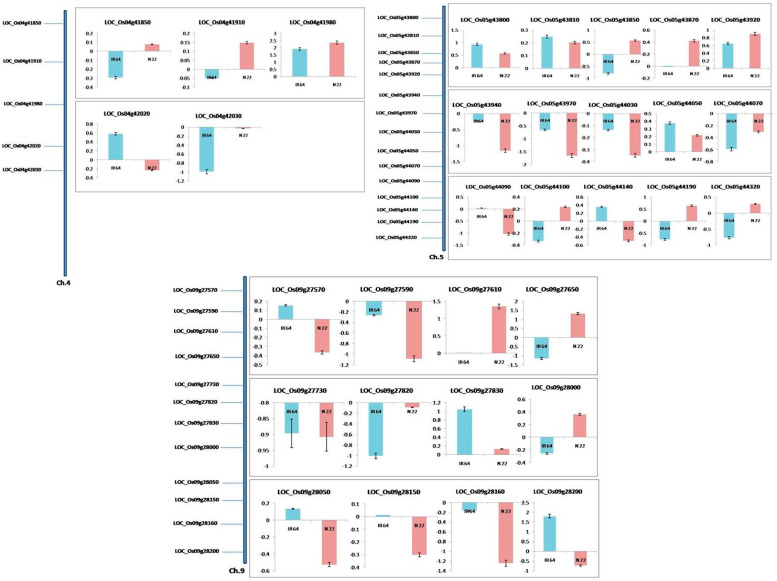
Expression profiling of the genes present in the selected QTLs regions which have single nucleotide polymorphism(s) in the parental genotypes, N22 and IR64, under eight days of heat stress by qPCR. The genes from the three QTL regions are shown in separate panels. The Y axis in all the graphs represents fold change in expression with respect to control (optimal temperature) conditions.

**Figure 7 plants-12-01697-f007:**
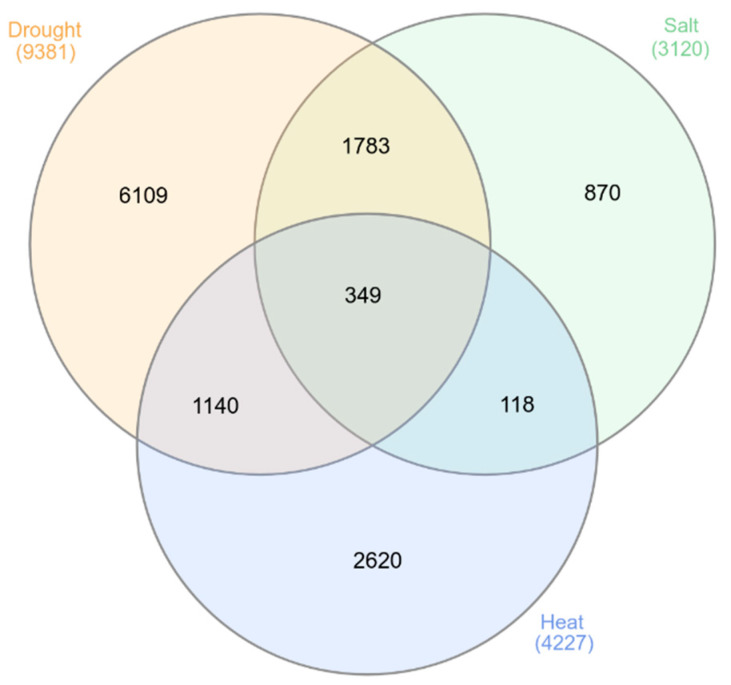
A Venn diagram representing the stress-responsive genes specific and common to drought, salinity and heat stress.

**Table 1 plants-12-01697-t001:** Details of the microarray datasets used for the identification of heat stress-responsive genes in rice.

S. No.	GEO ID	Variety	Tissue	Stage	Stress	No. of DEGs
1	GSE19983	Pusa Basmati	Seedlings	Seedling Stage	10 min	2024
					30 min	1313
2	GSE38665	Cultivar 996	Panicle	Reproductive Stage	20 min	2937
					1 h	3533
					2 h	3463
					4 h	4077
					8 h	3948
3	GSE55341	M202	Leaf	Vegetative Stage		2
4	GSE51426	Cultivar 996	Panicle	Reproductive Stage	2 h	1746
5	GSE45259		Flag Leaf		20 min	2951
					1 h	3651
					2 h	0
					4 h	2079
					8 h	4198
6	GSE1367464	Nagina 22 and IR64	Panicle		8 days	1104
					3 days heat	256
					1 day heat	1126
			Anther		3 days heat	126
7	GSE57154	Moroberekan	Pistil		1 day heat	435
			Panicle		8 days	1163

## Data Availability

The microarray data are available in the public domain as mentioned. The database is also available in the public domain. We have provided all other data either in the main article or as Appendix A.

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
