# Peer review of "Meta-Analysis of Microarray Data and Their Utility in Dissecting the Mapped QTLs for Heat Acclimation in Rice"

_plants, 2023, doi:10.3390/plants12081697_

Round 1

Reviewer 1 Report

In the manuscript entitled “A resource for identifying genes conferring high temperature stress tolerance in rice by merging global expression data and demonstration of its utility in dissecting the mapped QTLs for heat acclimation”, the authors constructed a database, RiceMetaSys-H, including 4227 heat stress responsive genes. Four candidate genes were identified for heat stress tolerance. My major concerns are as follows.

(1) Why did authors choose microarray datasets, but not RNA sequencing?

(2) What is the method to control false discovery rate?

(3) Four candidate genes should be verified by null mutations and/or overexpression lines.

Author Response

Response to Reviewer’s comments on the manuscript titled ‘A resource for identifying genes conferring high temperature stress tolerance in rice by merging global expression data and demonstration of its utility in dissecting the mapped QTLs for heat acclimation’ (Plants: 2117472)

Reviewer 1

Comment 1: Why did authors choose microarray datasets, but not RNA sequencing?

Response: In 2021, a comprehensive analysis of RNA-seq data under heat stress tolerance has been published with emphasis on alternative splicing (Vitoriano and Calixto, 2021, Plants). Further, in RNA-seq data, one need to do the analysis experiment by experiment (Vitoriano and Calixto, 2021, Plants) which makes meta-analysis not possible, at least to our knowledge. Most of the low abundant transcripts also need to be removed for normalization and this takes away the advantage of using the RNA-seq data. And we were confident about the established meta-analysis tools available in case of microarray data. That’s why we used only the microarray data. This portion is now added to the MS (Section 4, paragraph 2).

Comment 2: What is the method to control false discovery rate?

Response: We used Benjamini-Hochberg (BH) in the Limma tool to filter the differentially expressed genes. This controls the FDR which gives adjusted P value as output. This information is added in the methods section (Section 2.3).

Comment 3: Four candidate genes should be verified by null mutations and/or overexpression lines.

Response: Yes, in our laboratory, some of these genes are being validated by transgenic approach and this will take more time to get homozygous transgenic plants and then carry out their molecular and phenotype analysis. So this work is not a part of this manuscript.

Reviewer 2 Report

The title of the manuscript is lengthy, please make it short.

The abstract is too long, please make it succinct. It is worth mentioning the names of identified candidate genes in the abstract.

Keywords: Use keywords that are not there in the title.

What was the exact temperature set in the phenotyping facility for imposing heat stress? It is also important to mention the other growth conditions like photoperiod, soil moisture, and irrigation intervals.

Is there any reason for growing the control plants outside the phenotyping facility?

Section 2.3: Please reference all the bioinformatics tools used in section 2.3 for developing the RiceMetaSys-H database.

Please provide the weblink in the manuscript for RiceMetaSys-H database for users.

Section 2.4: Please provide enough background about three major QTLs, one each on chromosomes 4, 5, and 9.

Section 2.6: Please provide the list of primers in the supplementary information.

Section 3.1: Word transcriptome is misleading here, better to use microarray.

Author Response

Response to Reviewer’s comments on the manuscript titled ‘A resource for identifying genes conferring high temperature stress tolerance in rice by merging global expression data and demonstration of its utility in dissecting the mapped QTLs for heat acclimation’ (Plants: 2117472)

Reviewer 2

Comment 1: The title of the manuscript is lengthy, please make it short.

Response: The title of the manuscript has been modified as follows: ‘Meta-analysis of microarray data and its utility in dissecting the mapped QTLs for heat acclimation in rice’.

Comment 2: The abstract is too long, please make it succinct. It is worth mentioning the names of identified candidate genes in the abstract.

Response: The names of the four identified candidate genes are included in the abstract and we have also tried to make it brief (the word count has been brought down by 50 words).

Comment 3: Keywords: Use keywords that are not there in the title.

Response: Thank you for the suggestion. We have changed the title in the revised version and accordingly kept the following key words: Database for heat stress tolerance; stress-responsive genes; candidate genes; allele mining; QTL analysis.

Comment 4: What was the exact temperature set in the phenotyping facility for imposing heat stress? It is also important to mention the other growth conditions like photoperiod, soil moisture, and irrigation intervals.

Response: Keeping in view the diurnal variation, the temperature in the phenotyping facility was maintained 5oC higher than the atmospheric temperature. The highest temperature to which the plants were exposed was 39oC at afternoon (1:00 to 2:00 pm). The plants were grown in 16/8 photoperiod, adequate soil moisture was provided to the plants and care was taken not to expose them to moisture deficit. They were irrigated once a day preferably during morning (Section 2.1).

Comment 5: Is there any reason for growing the control plants outside the phenotyping facility?

Response: The phenotyping facility had only one chamber meant for growing rice with automated heat control and hence the control plants were not kept there.

Comment 6: Section 2.3: Please reference all the bioinformatics tools used in section 2.3 for developing the RiceMetaSys-H database.

Response: All the references related to bioinformatics tools are added and their sequence numbers have also been included in the text and accordingly the sequence numbers of the consecutive references have been updated.

Comment 7: Please provide the weblink in the manuscript for RiceMetaSys-H database for users.

Response: This was a mistake on our part not to include it earlier, the database URL is now added in the manuscript just after the keywords.

Comment 8: Section 2.4: Please provide enough background about three major QTLs, one each on chromosomes 4, 5, and 9.

Response: The information related to the three major QTLs have already been provided in section 3.3.1 and supplementary table 3 including their physical position and QTL interval.

Comment 9: Section 2.6: Please provide the list of primers in the supplementary information.

Response: The primer details are provided in supplementary table 6.

Comment10: Section 3.1: Word transcriptome is misleading here, better to use microarray.

Response: The word ‘transcriptome’ has been replaced with ‘microarray’.

Round 2

Reviewer 1 Report

The responses to my concerns are generally acceptable. I recommend the publication of the revised manuscript with ID “plants-2117472”.

Author Response

Thank you for accepting the revised version.